# Exploring anxiety symptoms, coping strategies, and socio-demographic influences among myocardial infarction patients in a tertiary care setting

Mohammad Sabeeh Ul Haq[1]☉*, Dua Nilofar Jawed[2]☉, Nehrish Patel[3]☉, Waqar Khan[4]‡, Aisha Alamgir[5]‡, Shagufta Yamin[6]‡, Rabia Anwar[7]‡, Tafazzul Hyder Zaidi[8], Munawar Khursheed[9]

1 Medical student (MBBS), Jinnah Sindh Medical University, Karachi, Pakistan, 2 Medical student (BSMT – RT), Jinnah Sindh Medical University, Karachi, Pakistan, 3 Senior registrar, National Institute of Cardiovascular Diseases (NICVD), Karachi, Pakistan, 4 Associate professor, National Institute of Cardiovascular Diseases (NICVD), Karachi, Pakistan, 5 Professor, Liaquat National Hospital and Medical College, Karachi, Pakistan, 6 Senior Lecturer, Department of Medical Education Sindh Institute of Medical Sciences, Sindh Institute of Urology and Transplantation (SIUT), Karachi, Pakistan, 7 DipCard PGR, National Institute of Cardiovascular Diseases (NICVD), Karachi, Pakistan, 8 Head of Department (HOD) Community Medicine, Jinnah Sindh Medical University, Karachi, Pakistan, 9 Professor Emergency Department, National Institute of Cardiovascular Diseases (NICVD), Karachi, Pakistan

☉ These authors contributed equally to this work.
‡ WK, AA, SY and RA also contributed equally to this work.
* sabeeh0404@gmail.com

## Abstract

### Objective

This study aimed to investigate levels of anxiety/anxiety symptoms and coping methods employed among patients suffering from myocardial infarction admitted in a tertiary care setting in a developing country. It also aimed to study the relationship of anxiety levels and coping strategies with each other and also the influence of various socio-demographic factors on anxiety levels and coping strategies used by the patients.

### Methods

A cross-sectional study was conducted at National Institute of Cardiovascular Disease (NICVD), Karachi, Pakistan involving 203 patients diagnosed with myocardial infarction. Sampling was conducted by non-probability convenience method. Patients were asked to fill a questionnaire comprising a socio-demographic form, HAM – A scale for anxiety symptoms and brief COPE scale for coping methods. Statistical analysis was performed using independent t-test, ANOVA, MANOVA, Pearson correlation and multiple regressions.

**Data availability statement:** The dataset for the study is available from Zenodo at https://doi.org/10.5281/zenodo.15238838.

**Funding:** The author(s) received no specific funding for this work.

**Competing interests:** The authors have declared that no competing interests exist.

## Results

Study comprised 203 patients (134 male, 69 female), 36.9% of the patients showed anywhere from moderate to severe levels of anxiety. Gender, history of myocardial infarction, family visits and hospital satisfaction were found to have statistical significance in influencing mean HAM – A scores among the population. Significant differences in problem focused coping was found among different gender and socio economic classes, while avoidance focused coping differences were found between patients receiving family visits and drug users as well as perceived hospital satisfaction. Hospital satisfaction also influenced emotional focused coping.

## Conclusion

This research highlights the importance of providing integrated psychological support for patients with myocardial infarction, specifically on anxiety and coping methods. By examining these factors, we can better understand how they influence mental health outcomes. Addressing anxiety and enhancing effective coping strategies are crucial for improving overall recovery and quality of life for these patients.

---

## Introduction

Anxiety is an experience often associated with a group of disorders termed collectively as "anxiety disorders" defined as a common mental health condition characterized by excessive worry, fear, or apprehension about future events, it can manifest through physical symptoms like restlessness, increased heart rate, and difficulty concentrating [1]. Anxiety can range from mild to severe and can significantly impact daily functioning. Coping strategies refer to the cognitive and behavioral strategies that individuals use to manage stress and anxiety. These methods can help people adapt to challenging situations, reduce emotional distress, and promote mental well-being [2]. There may be different sub categories of coping methods such as, problem- focused coping; involving directly finding solutions for the problem causing stress, emotional-focused coping; involves tackling emotional distress through methods such as mindfulness and social support, avoidance coping; using methods of avoidance and distraction, adaptive coping; using alternative strategies such as partaking in hobbies to alleviate stress, maladaptive coping; resorting to harmful activities/behaviors such as denial or drug abuse etc [3].

The hospital environment is one of naturally high stress, which contributes to anxiety among patients in hospital settings, due to various factors such as fear of diagnosis, treatment concerns, lack of support etc [4]. Studies in the past have highlighted the prevalence of anxiety in the hospital setting. In a study of 102 newly admitted patients in an acute care setting with low back pain, 44.1% and 31.4% were probable and possible anxiety cases, respectively, with a mean anxiety score of 19.46 measured using the Hospital Anxiety and Depression Scale (HADS) [5]. On a larger scale, a prospective cohort study of 2,064 cardiology patients showed mean

anxiety scores of 40.6 at intake, dropping to 19.5 at discharge, indicating how hospital experiences and treatment anticipation affect anxiety levels [6]. These studies collectively demonstrate that anxiety is a common issue in hospital settings, influenced by various health challenges and individual circumstances.

Anxiety is prevalent among myocardial infarction patients globally affecting their quality of life; in studies, lower socio-economic status has been shown to be associated with higher anxiety levels, and older patients of myocardial infarction suffering from anxiety show greater declines in quality of life, especially in the context of pain, discomfort and anxiety [7]. Long term psychological effects include increased distress in patients returning to work after suffering from a myocardial infarction [7]. Gender differences have also been demonstrated in the past with female patients showing statistically significant higher level levels of anxiety (p = 0.014) [8]. A meta-analysis of 12 studies confirmed that anxiety worsens prognosis among myocardial infarction patients (R = 1.27, p = <.001) [9]. Furthermore, some patients of myocardial infarction had reported death anxiety levels surpassing even that of cancer patients [10]. A cohort study showed that high anxiety scores were associated with a 1.31 times higher mortality rate of ten years [11]. Compared to non-myocardial infarction patients, those with the condition face significantly higher risks for anxiety (HR = 5.06) and depression (HR = 7.23) (where HR = Hazard ratio) [12]. This interplay between anxiety and physical health as demonstrated from the above studies emphasizes the need for integrated mental health support to help improve overall outcome and reduce recurrence of events among myocardial infarction patients.

Lifestyle factors are influenced by anxiety, as a U.S.-based survey revealed that 34.3% of myocardial infarction patients were physically inactive, with anxiety sensitivity linked to this inactivity [13]. A study on first-time myocardial infarction patients indicated that in-hospital medications and social relationship difficulties significantly correlate with anxiety levels, while self-perception of anxiety is a reliable indicator of actual anxiety [14]. The finding of anxiety among hospital patients carries over to Pakistan as well; a study in Pakistan found that 48.7% of 300 cancer patients had elevated anxiety or depression levels according to the Aga Khan University Anxiety and Depression Scale (AKUADS score > 20) [15]. Likewise pertaining to myocardial infarction patients in particular; In a Pakistani study, 27.24% of myocardial infarction patients exhibited depression, which signifies levels of psycho-social derangements that develop among patients of myocardial infarction [16].These studies collectively demonstrate that anxiety is a common issue in hospital settings, influenced by various health challenges and individual circumstances.

Given the evidence from above studies regarding anxiety among myocardial infarction patients and its various influencing factors it is reasonable to presume that understanding coping methods to alleviate anxiety is crucial. Studying the relationship/association between coping methods and anxiety and how they influence each other while considering several socio-demographic factors including age, family/social support, socio-economic class, number of myocardial infarctions, and treatment satisfaction can prove to have a potentially vital role in influencing patient's future wellness. Especially considering the unique context of a developing country and challenges that may be associated with a tertiary care setting. Pakistan being a developing country may not be as advanced when it comes to the understanding of mental health and thus it's influence on patient prognosis and betterment [17]. Additionally it is worth noting that there is a lack of literature on this topic for this patient population in particular. We aim to understand how different coping strategies affect anxiety in myocardial infarction patients. This can help us understand which strategies improve anxiety and which ones worsen it, ultimately improving both psychological and physiological outcomes. Furthermore, exploring coping and anxiety in myocardial infarction patients can enhance our understanding of the influence psychological well being has on cardiovascular health. In order to understand this influence it is important that we pinpoint effective interventions that address mental health among patients, which is often overlooked in clinical settings, particularly in resource-constrained environments [18]. This suggested influence of mental health on cardiovascular well-being is emphasized by previously mentioned literature that suggests psycho-social aspects not only influence mental well-being but also have a potential contribution to the prognosis of the cardiovascular disease at hand including events ranging from recurrent myocardial infarction all the way to mortality. This study would help conceptualize potential psychotherapeutic programs and interventions which

may serve to be useful for the particular patient population according to common themes of anxiety and coping strategies adopted by the individuals concerned. Thus, this investigation is particularly important and worthwhile given the literature and the pressing need for holistic care in the region [19].

## Materials and methods

A cross sectional study was conducted from July 21st 2024 to August 10th 2024 in the National Institute of Cardiovascular Diseases (NICVD), Karachi, Pakistan. The study was conducted on 203 patients of myocardial infarction admitted at the tertiary care hospital. The sample size was calculated using Open Epi 3.01 [20], by considering a total population based on estimated number of myocardial infarction patients that are admitted to the hospital within the time frame of data collection and using a confidence interval of 95%.

The data was collected by distributing a structured questionnaire. The sampling technique employed was non-probability convenience sampling. The questionnaire, which was originally developed in English, was translated into simple Urdu and then retranslated back into English with subsequent cross checking done in order to ensure its accuracy and validity, the final questionnaire was administered in Urdu, as doing such would ensure maximum comprehension among the selected sample. The questionnaire was divided into three main parts, the first part of the questionnaire mainly consisted of socio-demographic information including age, gender, number of previous myocardial infarctions, perceived family/social support, and hospital satisfaction. The second part consisted of the Hamilton Anxiety Rating Scale (HAM – A) [21] to assess anxiety levels and symptoms among the patients, whereas the third and final part of the questionnaire comprised of the Brief COPE scale [22–24] which was used to assess the coping strategies employed by the patients. These scales have been used extensively in past studies involving a variety of different patient populations including college students, stroke patients, hypertensive patients, chronic disease patients, cancer patients etc. [25–30]. Both HAM-A and brief COPE scales have been validated in Pakistan [31–32]

The questionnaire was given to the data collectors (authors of the study who did not currently work at NICVD) who distributed them among the myocardial infarction patients admitted at the hospital. In the light of convenience sampling participants were given the questionnaire upon recruitment without any delays. Written consent was taken from the participants and all ethical considerations and research protocols were observed. Data was collected in the form of pre-tested self-administered questionnaires. In order to standardize the questionnaires and for the purpose of examining content validity a pilot study was conducted among patients in NICVD that were in line with the criteria made for the sample population. These participants were not included in the final sample.

Data collected was analyzed using SPSS software version 20.0. Independent sample t- test, ANOVA and MANOVA were used to assess the relationship between socio-demographic factors and anxiety scores as well as socio demographic factors and coping strategies employed. Pearson correlation and multiple linear regressions were used to assess the association between anxiety scores and coping strategies employed. Continuous normally distributed data was described using means with standard deviation, whereas categorical data was described using frequencies. A significance level of $p < 0.05$ was considered. Normality of dependant variables was assessed using visual inspection of histograms, most of which displayed a relatively bell-shaped, symmetrical distribution, with minimal skewness of a few variables which were still largely symmetrical and near to normal distribution. Based on these visual assessments we assumed normality for the dependant variables in the subsequent analysis.

The study included patients 18 and older who were admitted to the hospital with myocardial infarction, who provided proper informed consent and understood the questionnaire provided to them. Patients under the age of 18 or patients with other cardiac conditions, lack of informed consent or those not able to comprehend the questionnaire due the cognitive impairment or any other reason were excluded from the study. Ethical approval of the study was obtained from the Institutional review board (IRB) at Jinnah Sindh Medical University (JSMU), as well as the departmental review committee (DRC) at National Institute of Cardiovascular Diseases (NICVD).

## Results

### General characteristics of sample population and socio-demographic distribution of brief COPE and HAM-A scores

Out of 203 sample patients included in the study 134 (66.0%) were male participants while 69 (34.0%) were female participants. Mean age of the sample population was 54 (SD ± 12.50). 32 (15.8%) of the patients were between ages 18–40, 114 (56.2%) of the patients were between ages 41–60, while 57 (28.1%) of the patients were between the ages of 61–85, we divided patients into age categories (18–40, 41–60, 61–85), in an attempt to reflect the typical classifications for young adults, middle-aged adults, and older adults. This enabled us to explore variability in coping style and anxiety across these different life stages, which can be relevant when studying the psychological consequences of myocardial infarction during early and late life stages. We selected the intervals to capture the unique challenges of these age groups. The minimum age of the sample was 18 while the maximum age was 85. 129 (63.5%) of the patients were experiencing their first incident of myocardial infarction whereas 74 (36.5%) of the patients had a prior history of myocardial infarction. Out of the patients with prior history of myocardial infarction 67 (33.0%) of them had 1–2 previous events of myocardial infarction whereas 7 (3.4%) of the patients had 3–4 previous events of myocardial infarction. No patient in the sample had more than 4 previous myocardial infarction events. In terms of socio-economic distribution of the sample 149 (73.4%) of the population fell into the lower socio-economic class while 49 (24.1%) were in the middle-socio-economic class and 5 (2.5%) were higher socio-economic class, it is important to note that this categorization of socio-economic status was based on household income of the patients with income of less than Rs 50,000 (Rupees PKR) being considered as lower socio-economic status, Rs 50,000–100,000 being considered as middle socio-economic status and greater than Rs 100,000 being considered as higher socio-economic status. A majority of the patients (87.4%) claimed that they had a support system (family and friends) and 119 (58.6%) of the patients reported frequent visits from family or friends. A large majority (88.2%) of the sample reported satisfaction with the healthcare that was being provided to them at the hospital. 36% of the sample population reported recreational drug use, this may be due to the diverse population served at NICVD, including low to middle income patients from Karachi and surrounding rural areas, where urban stress and easy access to drugs are common. Furthermore, NICVD has highly subsidized or often times free treatment which also influences the demographic patient population alongside playing a role in decreasing financial stress which may enable patients to prioritize other coping methods such as drug use.

There was a difference in HAM – A scores as well as brief COPE scores among the sample. For instance, males were found to have an average HAM – A score of 20.69 (± 8.62) whereas females had an average HAM – A score of 23.54 (± 9.11). Female population was found to have higher brief COPE scores in problem and emotional focused coping whereas males had higher scores in avoidance focused coping strategies. The mean HAM – A score also seemed to increase with increasing age group with the lowest average of 20.09 (± 7.21) among the 18–40 year old age group and the highest average of 22.63 (± 9.80) found among the oldest age group of 61–85. Problem focused brief COPE scores decreased with increasing age groups whereas avoidance focused brief COPE scores increased with the increasing age groups, emotional focused brief COPE scores were relatively similar between the age groups of 18–40 and 41–60 year olds but were slightly higher among the oldest age group of 61–85 year olds. Patients who had experienced previous myocardial infarctions had a higher average HAM – A score of 23.49 (± 9.91) as compared to those patients who were experiencing their first myocardial infarction who had an average HAM – A score of 20.60 (± 8.07). Patients with prior history of myocardial infarction demonstrated lower problem focused brief COPE scores and higher emotional and avoidance focused brief COPE scores compared to patients having their first myocardial infarction. Avoidance focused brief COPE scores also increased with increasing number of prior myocardial infarctions with the highest scores found in patients with 3–4 prior myocardial infarction events. The higher socio economic class demonstrated lower HAM – A score average than lower and middle income classes and also higher problem and emotional

focused brief COPE scores as well as lower avoidance focused brief COPE scores than the other two classes. Patients who were frequently visited by family and friends were found to have a lower HAM – A average of 19.98 (± 8.44) compared to those who did not receive visits that had an average of 24.02 (± 8.98). Both groups had similar problem and emotion focused brief COPE scores but the group that did not have visits from family and friends had higher scores for avoidance focused coping. Patients with recreational drug use had a higher average of avoidance focused coping, i.e., 2.05 (± 0.53) compared to patients without recreational drug use whose average score for avoidance focused coping was 1.68 (± 0.36). Patients who reported satisfaction to health care also demonstrated a lower average in HAM – A scores, with an average of 21.20 (± 8.87) compared to 25.08 (±8.26) among patients who were reportedly dissatisfied with the healthcare. Patients with health care satisfaction also demonstrated lower scores for problem, emotion as well as avoidance focused coping strategies. See Table 1.

**Table 1. Socio-demographic characteristics and socio-demographic distribution of coping and anxiety scores of the sample population.**

| | | Descriptive (Mean±SD or %) | HAM – A Scores (Mean±SD) | Problem Focused brief COPE Scores (Mean±SD) | Emotional focused brief COPE scores (Mean±SD) | Avoidant focused brief COPE scores (Mean±SD) |
|---|---|---|---|---|---|---|
| Gender | Male | 134 (66.0%) | 20.69 (± 8.62) | 2.01 (± 0.44) | 1.95 (± 0.38) | 1.85 (± 0..49) |
| | Female | 69 (34.0%) | 23.54 (± 9.11) | 2.17 (± 0.43) | 2.00 (± 0.39) | 1.75 (± 0.40) |
| Age (Mean±SD) (Age Range) | 18-40 | 32 (15.8%) | 20.09 (± 7.21) | 2.17 (± 0.48) | 1.95 (± 0.44) | 1.82 (± 0.53) |
| | 41-60 | 114 (56.2%) | 21.61 (± 8.81) | 2.08 (± 0.44) | 1.94 (± 0.32) | 1.77 (± 0.45) |
| | 61-85 | 57 (28.1%) | 22.63 (± 9.80) | 1.97 (± 0.40) | 2.03 (± 0.46) | 1.90 (± 0.45) |
| First MI | Yes | 129 (63.5%) | 20.60 (± 8.07) | 2.10 (± 0.46) | 1.95 (± 0.33) | 1.79 (± 0.47) |
| | No | 74 (36.5%) | 23.49 (± 9.91) | 2.00 (± 0.39) | 2.00 (± 0.46) | 1.84 (± 0.45) |
| Number of previous MI attacks (Mean±SD) | | 0.58 (± 0.88) | – | – | – | – |
| Number of previous MI attacks (Categorized) | 0 | 129 (63.5%) | 20.60 (± 8.07) | 2.10 (± 0.46) | 1.95 (± 0.33) | 1.79 (± 0.47) |
| | 1-2 | 67 (33.0%) | 23.55 (± 9.92) | 2.00 (± 0.36) | 2.00 (± 0.44) | 1.82 (± 0.44) |
| | 3-4 | 7 (3.4%) | 22.86 (± 10.54) | 1.92 (± 0.62) | 2.01 (± 0.65) | 2.07 (± 0.53) |
| Socio-economic status | Lower | 149 (73.4%) | 21.56 (± 8.88) | 2.05 (± 0.43) | 1.96 (± 0.36) | 1.79 (± 0.46) |
| | Middle | 49 (24.1%) | 22.33 (± 8.55) | 2.04 (± 0.44) | 1.95 (± 0.43) | 1.92 (± 0.45) |
| | Higher | 5 (2.5%) | 17.80 (± 12.43) | 2.70 (± 0.40) | 2.36 (± 0.32) | 1.37 (± 0.23) |
| Support system | Yes | 172 (87.4%) | 21.74 (± 8.78) | 2.08 (± 0.43) | 1.96 (± 0.37) | 1.81 (± 0.47) |
| | No | 31 (15.3%) | 21.19 (± 9.51) | 1.98 (± 0.50) | 2.02 (± 0.44) | 1.83 (± 0.42) |
| Visited by family and friends | Yes | 119 (58.6%) | 19.98 (± 8.44) | 2.06 (± 0.46) | 1.99 (± 0.41) | 1.76 (± 0.43) |
| | No | 84 (41.4%) | 24.02 (± 8.98) | 2.06 (± 0.41) | 1.93 (± 0.34) | 1.89 (± 0.50) |
| Recreational drug use | Yes | 73 (36.0%) | 21.00 (± 8.24) | 2.02 (± 0.42) | 1.92 (± 0.41) | 2.05 (± 0.53) |
| | No | 130 (64.0%) | 22.02 (± 9.22) | 2.09 (± 0.45) | 1.99 (± 0.36) | 1.68 (± 0.36) |
| Healthcare satisfaction | Yes | 179 (88.2%) | 21.20 (± 8.87) | 2.04 (± 0.43) | 1.95 (± 0.36) | 1.79 (± 0.45) |
| | No | 24 (11.8%) | 25.08 (± 8.26) | 2.22 (± 0.51) | 2.12 (± 0.49) | 2.00 (± 0.53) |
| Total | | 203 (100.0%) | 21.66 (± 8.87) | 2.06 (± 0.44) | 1.97 (± 0.38) | 1.81 (± 0.46) |

*Mean±Standard Deviation (SD) represents the average value±the variability in the dataset. This applies to continuous variables (HAM-A scores, brief COPE scores, mean number of previous myocardial infarction [MI] attacks)

*Percentages represent proportions, calculated as (part/total) × 100. This applies to categorical data or frequency-based variables (Gender distribution, age range distribution, proportion of patients having first MI, categorization of number of previous MI, categorization of socio-economic status, support system, visits by family and friends, recreational drug use, health care satisfaction, total population)

## Distribution of HAM-A and Brief COPE scores

Among the sample population 70 (34.5%) participants were found to have mild anxiety symptoms (indicated by HAM – A scores ranging from 0–17), 58 (28.6%) participants had mild to moderate anxiety symptoms (indicated by HAM – A scores ranging from 18–24), 39 (19.2%) of the participants had moderate to severe anxiety symptoms(indicated by HAM – A scores ranging from 25–30), and 36 (17.7%) participants had severe anxiety symptoms (indicated by HAM – A scores of 31 and greater). See Table 2.

Among problem focused coping strategies the highest average score of 2.31 (± 0.77) was found in the sub category of active coping while the lowest average score of 1.83 (± 0.63) was found in the subcategory of planning. For emotional focused coping strategies the highest average of 3.1 (± 0.95) was found in the subcategory of religious coping whereas the lowest average of 1.30 (± 0.48) was in the subcategory of humor, meanwhile in avoidance focused coping methods the highest average score was 2.24 (± 0.88) in the subcategory of self-distraction while the lowest being 1.62 (± 0.95) was in the subcategory of substance abuse. See Table 3.

## Relation between socio demographic characteristics with mean HAM-A scores and Brief COPE scores

Using T-test and ANOVA tests the difference in mean HAM – A scores were analyzed by dividing the sample into different socio-demographic groups. Male and female participants were found to have a mean difference in scores of -2.85 which

**Table 2. Distribution of HAM – A anxiety scores.**

|  | Mild anxiety –(HAM – A score of 0–17) | Mild – Moderate anxiety (HAM – A score of 18–24) | Moderate – Severe anxiety (HAM – A score of 25–30) | Severe anxiety (HAM – A score of 31+) |
|---|---|---|---|---|
| **Frequency(N)** | 70 | 58 | 39 | 36 |
| **Percentage(%)** | 34.48% | 28.57% | 19.21% | 17.73% |

*N represents the number of observations or the sample size in the specified category or group.*

*Percentage (%) represents the proportion of a given value relative to the total, calculated as (part/total) × 100*

**Table 3. Distribution of brief COPE scores.**

| Coping strategy | | Mean Score SD (±) |
|---|---|---|
| **Problem Focused Coping** 2.06 (± 0.44) | Active coping (PF) | 2.31(± 0.77) |
| | Informational support (PF) | 2.12 (± 0.79) |
| | Positive reframing (PF) | 1.99(± 0.71) |
| | Planning (PF) | 1.83 (± 0.63) |
| **Emotional Focused Coping** 1.97 (± 0.38) | Emotional Support (EF) | 2.08 (± 0.78) |
| | Venting (EF) | 1.61 (± 0.79) |
| | Humor (EF) | 1.3(± 0.48) |
| | Acceptance (EF) | 1.96 (± 0.69) |
| | Religion (EF) | 3.1 (± 0.95) |
| | Self blame (EF) | 1.7 (± 0.77) |
| **Avoidance Focused Coping** 1.81 (± 0.46) | Self distraction (AF) | 2.24 (± 0.88) |
| | Denial (AF) | 1.68 (± 0.81) |
| | Substance abuse (AF) | 1.62 (± 0.95) |
| | Behavioral disengagement (AF) | 1.71 (± 0.70) |

*SD (±) represents the Standard Deviation, which measures the amount of variability or dispersion from the mean in the dataset.*

was statistically significant (p = 0.03, t = -2.18), The difference in mean HAM – A scores between patients who were experiencing their first myocardial infarction with those who had prior history of myocardial infarction was of -2.88 and this was also found to be statistically significant (p = 0.026, t = -2.88). The difference in mean HAM – A scores between patients who received visits from family and friends against those who didn't was –4.04 and this difference was statistically significant (p = 0.001, t = -3.27), likewise a statistically significant difference in means measuring to -3.88 (p = 0.044, t = -2.03) was found among patients who reported hospital satisfaction against those that did not. No statistically significant difference was found between mean HAM – A scores in patients of varying age groups, number of previous myocardial infarction or different socio-economic distribution. See Table 4 and 5.

Using MANOVA the difference in mean brief COPE scores including problem, emotion and avoidant focused coping methods was analyzed between different socio-demographic distributions of the patients population to find any significant differences. It was found that between male and female population the difference in mean problem-focused brief COPE scores was statistically significant (p = 0.013, F = 6.28, $\eta_p^2 = 0.030$), no statistically significant difference was found between emotion and avoidant focused brief COPE scores. A statistically significant difference in problem focused brief COPE scores was also found among patients of different socio economic classes (p = 0.005, F = 5.381, $\eta_p^2 = 0.051$), varying socio-economic classes also showed statistically significant difference in their avoidant focused brief COPE scores (p = 0.022, F = 3.868, $\eta_p^2 = 0.037$). Among patients that received visits from family and friends and those who did not there was a statistically significant difference in mean avoidant focused brief COPE scores (p = 0.046, F = 4.041, $\eta_p^2 = 0.020$), similarly between drug users and non drug users there was also a statistically significant difference in avoidant focused brief COPE scores (p = 0.000, F = 35.686, $\eta_p^2 = 0.151$). In patients who reported hospital satisfaction against those who did not there was statistical significance in mean emotional focused coping (p = 0.035, F = 4.500, $\eta_p^2 = 0.022$) as well as avoidant focused coping (p = 0.042, F = 4.183, $\eta_p^2 = 0.020$). See Table 6.

**Table 4. Relation between socio-demographic characteristics and mean HAM – A scores for anxiety (using independent sample t – test).**

| | | N | Mean HAM – A Scores | ± SD | ±SE Mean | Mean Difference | T | df | P |
|---|---|---|---|---|---|---|---|---|---|
| Gender | Male | 134 | 20.69 | ± 8.62 | ± 0.74 | - 2.85 | -2.18 | 201 | **0.03*** |
| | Female | 69 | 23.54 | ± 9.11 | ± 1.09 | | | | |
| First MI | Yes | 129 | 20.60 | ± 8.07 | ± 0.71 | -2.88 | -2.24 | 201 | **0.026*** |
| | No | 74 | 23.49 | ± 9.91 | ± 1.15 | | | | |
| Support system | Yes | 172 | 21.74 | ± 8.78 | ± 0.67 | 0.54 | 0.314 | 201 | 0.754 |
| | No | 31 | 21.19 | ± 9.51 | ± 1.70 | | | | |
| Visited by family and friends? | Yes | 119 | 19.98 | ± 8.44 | ± 0.77 | -4.04 | -3.27 | 201 | **0.001*** |
| | No | 84 | 24.02 | ± 8.98 | ± 0.98 | | | | |
| Drug use | Yes | 73 | 21.00 | ± 8.24 | ± 0.96 | -1.02 | -0.78 | 201 | 0.432 |
| | No | 130 | 22.02 | ± 9.22 | ± 0.80 | | | | |
| Hospital satisfaction | Yes | 179 | 21.20 | ± 8.87 | ± 0.66 | -3.88 | -2.03 | 201 | **0.044*** |
| | No | 24 | 25.8 | ± 8.26 | ± 1.68 | | | | |

*N represents the number of observations or the sample size in the specified category or group.

*SD (±) represents the Standard Deviation, which measures the amount of variability or dispersion from the mean in the dataset.

*±SE Mean represents the Standard Error of the Mean, which indicates the variability of the sample mean estimate from the true population mean.

*t represents the test statistic from the t-test, which is used to determine whether there is a significant difference between the means of two groups.

*df (degrees of freedom) represents the number of independent values that can vary in the calculation of a statistic, often used to determine the appropriate distribution for hypothesis testing.

*p represents the p-value, which tests the null hypothesis. A p-value of ≤ 0.05 indicates statistical significance, while a p-value > 0.05 suggests that the result is not statistically significant.

**Table 5. Relation between socio-demographic characteristics and mean HAM-A scores for anxiety (using ANOVA).**

| | | N | Mean HAM – A Score | ± SD | ± SE Mean | df between groups | df within groups | F | p |
|---|---|---|---|---|---|---|---|---|---|
| **Age range** | 18 - 40 | 32 | 20.09 | ± 7.21 | ± 1.27 | 2 | 200 | 0.841 | 0.433 |
| | 41 - 60 | 114 | 21.61 | ± 8.81 | ± 0.82 | | | | |
| | 61 - 85 | 57 | 22.63 | ± 9.80 | ± 1.29 | | | | |
| **Number of previous MI** | 0 | 129 | 20.60 | ± 8.07 | ± 0.71 | 2 | 200 | 2.536 | 0.082 |
| | 1 - 2 | 67 | 23.55 | ± 9.92 | ± 1.21 | | | | |
| | 3 - 4 | 7 | 22.86 | ± 10.54 | ± 3.98 | | | | |
| **Socio-economic status** | Lower | 149 | 21.56 | ± 8.88 | ± 0.72 | 2 | 200 | 0.617 | 0.540 |
| | Middle | 49 | 22.33 | ± 8.55 | ± 1.22 | | | | |
| | Higher | 5 | 17.80 | ± 12.43 | ± 5.56 | | | | |

*For definitions of N, SD, SE Mean, df, and p, please refer to the footnotes in Table 4.*

*F represents the F-statistic from the ANOVA test, which compares the variance between groups to the variance within groups. A larger F-value suggests a greater difference between group means relative to the variability within the groups, helping to determine whether the means of different groups are significantly different.*

### Relation between coping methods and HAM-A anxiety scores

When using Pearson's correlation between total HAM – A scores and mean brief COPE scores for problem, emotional and avoidant focused coping separately. A weakly positive and significant correlation was found between total HAM – A scores and the mean scores for coping strategies. See Table 7.

**Multiple regression model 1.** A multiple regression model was produced using subcategories of problem focused coping as independent variables and total HAM – A scores as the dependant variable. It was found that a positive relation existed between positive reframing and total HAM – A scores (B = 3.324, t = 3.706, p = 0.000). No significance was found among other subcategories of problem focused coping. The first model accounted for 9% of the variance in total HAM – A scores. See Table 8.

**Multiple regression model 2.** A multiple regression model was produced using subcategories of emotional focused coping as the independent variables and total HAM – A scores as the dependant variable. A positive significant relation was found between venting and total HAM – A scores (B = 2.545, t = 2.980, p = 0.003), and religious coping and total HAM – A scores (B = 1.780, t = 2.887, p = 0.004), meanwhile a negative significant relation was found between acceptance and total HAM – A scores (B = -3.283, t = -3.679, p = 0.000). This model accounted for 16% of the variance in total HAM – A scores. See Table 9.

**Multiple regression model 3.** A multiple regression model was produced using subcategories of avoidant focused coping as independent variables and total HAM – A scores as the dependant variable. A significant and positive relation was found between denial and total HAM – A scores (B = 1.668, t = 2.294, p = 0.023) and also between behavioral disengagement and total HAM – A scores (B = 3.934, t = 4.592, p = 0.000). This model accounted for 13.6% of the variance in total HAM – A scores. See Table 10.

### Discussion

In our study, based on HAM-A scores, 36.4% of patients experienced moderate to severe anxiety. It shows that anxiety is common among myocardial infarction patients. Significant differences in HAM-A scores were found across various socio-demographic groups, including gender, history of myocardial infarction, frequency of visits from family and friends, and satisfaction with healthcare services. This suggests that anxiety levels are influenced by multiple patient factors beyond the heart attack itself.

**Table 6. Relation between socio-demographic factors and brief cope scores (using MANOVA).**

| Coping strategies &Variables | | N | Mean brief COPE Score | ± SD | Df | F | p | $\eta_p^2$ |
|---|---|---|---|---|---|---|---|---|
| **Gender** | | | | | | | | |
| **Problem Focused Coping** | Male | 134 | 2.01 | ± 0.44 | 1 | 6.28 | **0.013*** | 0.03 |
| | Female | 69 | 2.17 | ± 0.43 | | | | |
| **Emotional Focused Coping** | Male | 134 | 1.95 | ± 0.38 | 1 | 0.8 | 0.372 | 0.004 |
| | Female | 69 | 2 | ± 0.29 | | | | |
| **Avoidance Focused Coping** | Male | 134 | 1.85 | ± 0.49 | 1 | 2.08 | 0.151 | 0.01 |
| | Female | 69 | 1.75 | ± 0.40 | | | | |
| **Age range** | | | | | | | | |
| **Problem Focused Coping** | 18 - 40 | 32 | 2.17 | ± 0.48 | 2 | 2.522 | 0.083 | 0.025 |
| | 41 - 60 | 114 | 2.08 | ± 0.44 | | | | |
| | 60 - 85 | 57 | 1.96 | ± 0.40 | | | | |
| **Emotional Focused Coping** | 18 - 40 | 32 | 1.95 | ± 0.44 | 2 | 1.101 | 0.335 | 0.011 |
| | 41 - 60 | 114 | 1.94 | ± 0.32 | | | | |
| | 61 - 85 | 57 | 2.03 | ± 0.46 | | | | |
| **Avoidance Focused Coping** | 18 - 40 | 32 | 1.82 | ± 0.53 | 2 | 1.593 | 0.206 | 0.016 |
| | 41 - 60 | 114 | 1.77 | ± 0.45 | | | | |
| | 61 - 85 | 57 | 1.9 | ± 0.45 | | | | |
| **First MI** | | | | | | | | |
| **Problem Focused Coping** | Yes | 129 | 2.1 | ± 0.46 | 1 | 2.706 | 0.102 | 0.013 |
| | No | 74 | 2 | ± 0.39 | | | | |
| **Emotional Focused Coping** | Yes | 129 | 1.95 | ± 0.33 | 1 | 0.992 | 0.321 | 0.005 |
| | No | 74 | 2 | ± 0.46 | | | | |
| **Avoidance Focused Coping** | Yes | 129 | 1.79 | ± 0.47 | 1 | 0.539 | 0.464 | 0.003 |
| | No | 74 | 1.84 | ± 0.45 | | | | |
| **Number of previous MI** | | | | | | | | |
| **Problem Focused Coping** | 0 | 129 | 2.1 | ± 0.46 | 2 | 1.447 | 0.238 | 0.014 |
| | 1 - 2 | 67 | 2 | ± 0.36 | | | | |
| | 2 - 3 | 7 | 1.92 | ± 0.62 | | | | |
| **Emotional Focused Coping** | 0 | 129 | 1.95 | ± 0.33 | 2 | 0.494 | 0.611 | 0.005 |
| | 1 - 2 | 67 | 2 | ± 0.44 | | | | |
| | 2 - 3 | 7 | 2.01 | ± 0.65 | | | | |
| **Avoidance Focused Coping** | 0 | 129 | | ± 0.47 | 2 | 1.138 | 0.323 | 0.011 |
| | 1 - 2 | 67 | | ± 0.44 | | | | |
| | 2 - 3 | 7 | | ± 0.53 | | | | |
| **Socio-economic status** | | | | | | | | |
| **Problem Focused Coping** | Lower | 149 | 2.05 | ± 0.43 | 2 | 5.381 | **0.005*** | 0.051 |
| | Middle | 49 | 2.04 | ± 0.44 | | | | |
| | Higher | 5 | 2.7 | ± 0.40 | | | | |
| **Emotional Focused Coping** | Lower | 149 | 1.96 | ± 0.36 | 2 | 2.701 | 0.07 | 0.026 |
| | Middle | 49 | 1.95 | ± 0.43 | | | | |
| | Higher | 5 | 2.36 | ±0.32 | | | | |
| **Avoidance Focused Coping** | Lower | 149 | 1.79 | ± 0.46 | 2 | 3.868 | **0.022*** | 0.037 |
| | Middle | 49 | 1.92 | ± 0.45 | | | | |
| | Higher | 5 | 1.37 | ± 0.23 | | | | |

*(Continued)*

**Table 6.** (Continued)

| Coping strategies &Variables | | N | Mean brief COPE Score | ± SD | Df | F | p | $\eta_p^2$ |
|---|---|---|---|---|---|---|---|---|
| **Gender** | | | | | | | | |
| **Support system** | | | | | | | | |
| **Problem Focused Coping** | Yes | 172 | 2.08 | ± 0.43 | 1 | 1.422 | 0.234 | 0.007 |
| | No | 31 | 1.97 | ± 0.50 | | | | |
| **Emotional Focused Coping** | Yes | 172 | 1.96 | ± 0.37 | 1 | 0.803 | 0.371 | 0.004 |
| | No | 31 | 2.02 | ± 0.44 | | | | |
| **Avoidance Focused Coping** | Yes | 172 | 1.81 | ± 0.47 | 1 | 0.073 | 0.787 | 0 |
| | No | 31 | 1.83 | ± 0.42 | | | | |
| **Visited by Family and Friends** | | | | | | | | |
| **Problem Focused Coping** | Yes | 119 | 2.06 | ± 0.46 | 1 | 0 | 0.984 | 0 |
| | No | 84 | 2.06 | ± 0.41 | | | | |
| **Emotional Focused Coping** | Yes | 119 | 1.99 | ± 0.41 | 1 | 1.418 | 0.235 | 0.007 |
| | No | 84 | 1.93 | ± 0.34 | | | | |
| **Avoidance Focused Coping** | Yes | 119 | 1.76 | ± 0.43 | 1 | 4.041 | **0.046*** | 0.02 |
| | No | 84 | 1.89 | ± 0.50 | | | | |
| **Drug use** | | | | | | | | |
| **Problem Focused Coping** | Yes | 73 | 2.02 | ± 0.42 | 1 | 0.931 | 0.336 | 0.005 |
| | No | 130 | 2.09 | ± 0.45 | | | | |
| **Emotional Focused Coping** | Yes | 73 | 1.92 | ± 0.41 | 1 | 1.548 | 0.215 | 0.008 |
| | No | 130 | 1.99 | ± 0.36 | | | | |
| **Avoidance Focused Coping** | Yes | 73 | 2.05 | ± 0.53 | 1 | 35.686 | **0.000*** | 0.151 |
| | No | 130 | 1.68 | ± 0.36 | | | | |
| **Hospital Satisfaction** | | | | | | | | |
| **Problem Focused Coping** | Yes | 179 | 2.04 | ± 0.43 | 1 | 3.375 | 0.068 | 0.017 |
| | No | 24 | 2.22 | ± 0.51 | | | | |
| **Emotional Focused Coping** | Yes | 179 | 1.95 | ± 0.36 | 1 | 4.5 | **0.035*** | 0.022 |
| | No | 24 | 2.12 | ± 0.49 | | | | |
| **Avoidance Focused Coping** | Yes | 179 | 1.79 | ± 0.45 | 1 | 4.183 | **0.042*** | 0.02 |
| | No | 24 | 2 | ± 0.53 | | | | |

*For definitions of N, SD, SE Mean, df, and p, please refer to the footnotes in Table 4.

*ηp² (partial eta-squared) represents the proportion of the variance in the dependent variable explained by a particular independent variable, after accounting for other variables. A higher value indicates a larger effect size.

The variation in HAM-A scores in our study is aligned with past research using different anxiety assessment tools. Problem-focused brief COPE scores showed a significant difference between men and women, with women scoring slightly higher. This suggests that female patients tend to take action-based coping steps like problem-solving and planning. Similarly results were found for the patients from higher-income backgrounds had higher problem-focused coping scores, meaning they were more likely to use problem-solving strategies. However, the smaller sample size for this group may have affected the results. This distribution is expected, since this study was conducted in a tertiary care hospital in Pakistan, a low resource and low income country.

Avoidant-focused coping was affected by socioeconomic status, with higher-income patients showing the lowest scores. This suggests that patients with good socioeconomic status avoid escape, while middle-income patients, who had the highest avoidant-focused coping scores, may be more prone to such behaviors. Also, patients whose family and friends pays visits to the patient were less likely to use avoidance as a coping mechanism this makes sense, as family

**Table 7. Relation between coping strategy scores and total ham – a scores (using pearson's correlation).**

|  | Total HAM – A Scores | Total HAM – A Scores | Total HAM – A Scores |
| --- | --- | --- | --- |
|  | N | r | P |
| Problem Focused Coping | 203 | 0.139 | **0.049*** |
| Emotional Focused Coping | 203 | 0.181 | **0.010*** |
| Avoidance Focused Coping | 203 | 0.162 | **0.021*** |

*For definitions of N and p, please refer to footnotes in table 4.

* r represents the Pearson correlation coefficient, which measures the strength and direction of a linear relationship between two continuous variables. Values range from -1 (perfect negative correlation) to +1 (perfect positive correlation), with 0 indicating no correlation.

**Table 8. Multiple regression model 1.**

| Coping Mechanism (PF) | B (Regression Weight) | t | p-value |
| --- | --- | --- | --- |
| Active Coping | 0.346 | 0.403 | 0.687 |
| Informational Support | 1.085 | 1.292 | 0.198 |
| Positive Reframing | 3.324 | 3.706 | 0.000* |
| Planning | -1.509 | -1.383 | 0.168 |
| **Overall Model Fit** |  |  |  |
| R (Multiple Correlation) | 0.301 |  |  |
| R² (Variance Explained) | 0.09 |  |  |
| F-statistic | 4.915 |  |  |

* B represents the unstandardized regression coefficient, which indicates the change in the dependent variable for a one-unit increase in the independent variable, while holding other variables constant.

* t represents the t-statistic, which tests whether the regression coefficient (**B**) is significantly different from zero. A positive t-value indicates a positive relationship between the independent and dependent variables, while a negative t-value suggests a negative relationship. A larger **t** value (in absolute terms) suggests a stronger predictor, whereas a smaller **t** value suggests a weaker predictor.

* p represents the p-value, which tests the null hypothesis for each regression coefficient (**B**) in the model. A p-value ≤ 0.05 indicates that the independent variable is a significant predictor of the dependent variable, while a p-value > 0.05 suggests it is not.

**Table 9. Multiple regression model 2.**

| Coping Mechanism (EF) | B (Regression Weight) | t | p-value |
| --- | --- | --- | --- |
| Emotional Support | 0.636 | 0.798 | 0.426 |
| Venting | 2.545 | 2.98 | 0.003* |
| Humor | 1.534 | 1.21 | 0.228 |
| Acceptance | -3.283 | -3.679 | 0.000* |
| Religion | 1.78 | 2.887 | 0.004* |
| Self-Blame | 0.64 | 0.707 | 0.48 |
| **Overall Model Fit** |  |  |  |
| R (Multiple Correlation) | 0.4 |  |  |
| R² (Variance Explained) | 0.16 |  |  |
| F-statistic | 6.211 |  |  |

* For definitions of **B**, **t**, and **p-value**, please refer to the footnotes in Table 8.

and friends provide emotional support and will power, making patients feel stronger and reducing avoidance behaviors. Patients who use drugs are more likely to use avoidant coping, which is expected since substance use is often a way to escape reality.

**Table 10. Multiple regression model 3.**

| Coping Mechanism (AF) | B (Regression Weight) | t | p-value |
|---|---|---|---|
| Self-Distraction | -0.166 | -0.244 | 0.807 |
| Denial | 1.668 | 2.294 | 0.023* |
| Substance Use | -0.942 | -1.473 | 0.142 |
| Behavioral Disengagement | 3.934 | 4.592 | 0.000* |
| Overall Model Fit | | | |
| R (Multiple Correlation) | 0.369 | | |
| R² (Variance Explained) | 0.136 | | |
| F-statistic | 7.818 | | |

*\* For definitions of **B**, **t**, and **p-value**, please refer to the footnotes in Table 8.*

Healthcare satisfaction also affected coping strategies. Patients satisfied with their care were less likely to use avoidant-focused coping than those dissatisfied. From this we may assume that a good healthcare experience encourages patients to engage in treatment and use problem-focused coping. On the other hand according to the results, dissatisfaction with healthcare can possibly lead to frustration, helplessness, or distrust, increasing avoidant coping. Patients can have a proactive mindset if he has a feeling of support and value, while dissatisfaction can push people toward avoidance. Patients satisfied with their healthcare were also less likely to use emotion-focused coping, possibly because trust in healthcare helps reduce emotional distress.

There were weak positive links between all three coping strategies and total HAM-A scores, showing a minor but present connection between coping methods and anxiety levels. Higher HAM-A scores were linked to coping strategies like positive reframing, venting, religion, denial, and behavioral disengagement. In contrast, patients who accepted their condition had lower HAM-A scores.

Studies have shown differences in how anxiety and coping methods affect myocardial infarction patients. Cognitive and Dialectical Behavioral Therapy has helped shift patients from emotional to problem-focused coping, improving stress management and health outcomes [33]. Older patients rely more on spiritual coping, while men generally have lower social and spiritual support. Also, those patients who have more chronic myocardial infarction receive less social support. Anxiety and depression negatively affect coping, highlighting the need for mental health support in rehabilitation [34]. Regarding disease outcomes, studies have shown that emotion-focused coping strategies improve left ventricular ejection fraction, suggesting that strengthening these coping strategies can help patients recover after a heart attack [35]. Positive coping methods like planning and seeking emotional support build resilience, while negative ones like denial and venting slow recovery [36]. The link between coping strategies, anxiety, and depression highlights the need to include coping assessments in clinical practice to support better mental and physical recovery [37]. Research also suggests that optimistic coping strategies work best for myocardial infarction patients, while emotional and palliative strategies are used less and are less effective. This highlights the need for personalized, culturally sensitive patient care, promoting positive coping while addressing barriers to better strategies [38]. Certain patterns of behavior from our findings were also common with past studies such as higher problem focused coping among individuals with higher incomes [39], decreased use of avoidant coping among patients having frequent family visits [40], as well as increased avoidant coping behaviors among individuals with history of drug use [41]. In the past good healthcare has also shown to increase problem-focused coping tendencies [42], while a poor healthcare satisfaction has been shown to have an association with avoidance related behaviors [43,44]. The expression of acceptance has also been shown to be associated with decreased anxiety [45].

This study provides a comprehensive analysis in regards to the relationship between anxiety and coping methods among myocardial infarction patients, while providing novelty in regards to investigating this relationship in a

resource-limited setting especially Pakistan. With the use of a wide array of statistical methods we have been able to help understand how socio-demographic factors like gender, socio-economic status, social support, drug abuse, hospital satisfaction, and past history of myocardial infarction may influence both anxiety and coping methods among the patients. Additionally the large sample size of 203 helps increase the reliability of the study making it more applicable to a broader population with similar socio-economic standing. The collective literature mentioned in this study along with the study's own findings also help to emphasize the significance of anxiety in myocardial infarction patients, as well as the impact of coping strategies on their psychological health, which highlights the need for personalized interventions in caring for such patients.

It is also important to acknowledge the limitations of this study. Firstly, the $R^2$ values observed in the regression models are relatively low at 9%, 16% and 13.6% for models one, two and three respectively. Although this doesn't reflect a poor or ineffective model, it is important to consider that such results suggest that the model only accounts for a small proportion of the variance in anxiety levels (i.e., HAM-A scores). This could mean that certain factors such as pre-existing mental health problems, medication use, personal affairs, and other socio-demographic variables which were not included in the models, could possibly account for the remaining unexplored variance in HAM-A scores. Additionally the cross-sectional design of the study limits the ability to establish causality between coping strategies and anxiety, as it does not capture the changes over time. It is also worthwhile to consider that since the data is self-reported by patients, the possibility of response bias exists which may affect the accuracy of some of the responses. Since the study is conducted on a relatively distinct population, i.e., patients of a tertiary care hospital in a developing country, with majority patients being of lower-socioeconomic background seeking inexpensive care we should acknowledge the fact that such results may not be as applicable in more affluent settings, or in healthcare systems with different economic and cultural contexts.

## Conclusion

In conclusion, this study details the significant prevalence of anxiety among myocardial infarction patients and highlights the associated factors relating to anxiety and coping strategies implemented. The findings of this study can be used as a guideline in future research to help patients in improving their psychological well-being. The findings indicate that tailored interventions addressing individual differences, such as gender, history of myocardial infarction, and social support, are essential for effectively managing anxiety. By fostering adaptive coping mechanisms, particularly acceptance and problem-focused strategies, healthcare providers can enhance patients' resilience and improve overall recovery outcomes. This research emphasizes the need for integrated psychological support in cardiac care, ultimately aiming to optimize mental health for myocardial infarction patients.

## Acknowledgments

We would like to acknowledge the patients of the hospital who's cooperation made this study possible as well as the National Institute of Cardiovascular Disease for hosting this study.

## Author contributions

**Conceptualization:** Mohammad Sabeeh Ul Haq, Dua Nilofar Jawed, Nehrish Patel, Waqar Khan, Shagufta Yamin, Tafazzul Hyder Zaidi.

**Data curation:** Mohammad Sabeeh Ul Haq, Dua Nilofar Jawed.

**Formal analysis:** Mohammad Sabeeh Ul Haq, Aisha Alamgir.

**Investigation:** Mohammad Sabeeh Ul Haq, Dua Nilofar Jawed, Nehrish Patel, Rabia Anwar, Munawar Khursheed.

**Methodology:** Mohammad Sabeeh Ul Haq, Dua Nilofar Jawed, Nehrish Patel.

**Visualization:** Mohammad Sabeeh Ul Haq.

**Writing – original draft:** Mohammad Sabeeh Ul Haq, Dua Nilofar Jawed, Rabia Anwar.

**Writing – review & editing:** Nehrish Patel, Waqar Khan, Aisha Alamgir, Shagufta Yamin, Tafazzul Hyder Zaidi, Munawar Khursheed.

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
