## [Decision Letter · Decision Letter 0]

8 Jan 2025

PONE-D-24-52101Exploring anxiety symptoms, coping strategies, and socio-demographic influences among myocardial infarction patients in a tertiary care setting.PLOS ONE

Dear Dr. Ul Haq,

Thank you for submitting your manuscript to PLOS ONE. After careful consideration, we feel that it has merit but does not fully meet PLOS ONE’s publication criteria as it currently stands. Therefore, we invite you to submit a revised version of the manuscript that addresses the points raised during the review process.

We look forward to receiving your revised manuscript.

Kind regards,

Pasyodun Koralage Buddhika Mahesh

Academic Editor

PLOS ONE

Reviewers' comments:

Reviewer's Responses to Questions

**Comments to the Author**

1. Is the manuscript technically sound, and do the data support the conclusions?

Reviewer #1: Partly

Reviewer #2: No

2. Has the statistical analysis been performed appropriately and rigorously? 

Reviewer #1: I Don't Know

Reviewer #2: No

3. Have the authors made all data underlying the findings in their manuscript fully available?

Reviewer #1: Yes

Reviewer #2: No

4. Is the manuscript presented in an intelligible fashion and written in standard English?

Reviewer #1: Yes

Reviewer #2: Yes

5. Review Comments to the Author

Reviewer #1: Apparently there are some methodological issues that interfere with the study results and conclusions. I wonder whether the assumptions are met with the statistical tests performed. Authors need to indicate how statistical test assumptions are met.

Need to mention whether the anxiety assessment tools are validated for the study setting (or the country).

Need to mention about the study limitations.

Reviewer #2: Thank you for submitting a research manuscript on a highly relevant topic: patients with myocardial infarctions, their anxiety, and coping skills. However, there are significant revisions needed to enhance the quality of the manuscript. These revisions are outlined below.

It is important to note that commenting on this manuscript is challenging due to the absence of line numbers and the line spacing not adhering to PLOS One manuscript guidelines.

Major revisions

1.Introduction :

The authors should pay closer attention to properly citing the literature used throughout the manuscript as not citing sources properly is considered plagiarism. E.g.

• In the first paragraph - The first sentence (the source for the definition of anxiety) needs a citation

• The second and fourth sentences also need to be cited

• In the second para – The first sentence needs a citation

• In the third para – the second and third statements are these from the same article as [5]?

• In the 6th para (before methods) etc.

In the presentation of the literature, it would be most effective to structure it in a manner that describes international evidence, followed by regional evidence, and then concluding with local evidence. This organization will enhance clarity and understanding.

2.Materials and Methods:

Second para – second sentence – So, was the final questionnaire administered in English or Urdu?

Has the HAM tool or the COPE scale previously been validated in Pakistan? If not, the authors should at least describe the cultural adaptation process of the tools.

Third para- the last sentence is not clear- Is it the same participants or the same sample population? Using the same participants to pre-test the questionnaire will invariably introduce bias.

Who were the data collectors? Health staff? Using healthcare workers to conduct data collection, especially when it involves questions about the satisfaction of services, does not seem ideal as it will induce bias.

The authors have used non-probability convenient sampling to capture the sample size of 203 individuals. However, the whole analysis depended on the assumption that the data is normally distributed. It is important to note that data from non-probability samples may not fit the assumptions of parametric tests such as the t-test. Looking at the data, gives an impression that the data is skewed. As the author guidelines state that the “authors must describe any analysis carried out to confirm the data meets the assumptions of the analysis performed (e.g. linearity, co-linearity, normality of the distribution)” suggests to include how the authors assumed that the data was normally distributed.

3. Results:

It would be better if the authors could explain the basis of categorizing the age into three categories 18-40, 41-60 and 61-85. The usual practice is to divide into age groups with equal intervals.

The participants were divided into lower, middle and higher socio-economic status. It would be beneficial for non-local readers of the manuscript to understand what these categories represent. Is it based on income? Assets or Education levels?

Tables: In column headings, first column, better to keep only the frequency and percentage and the Mean and SD have already been described in the text. No need for repetition.

Discussion

It's always better to summarise the findings of your research first and then compare it with available research evidence.

On all three regression models, the R2 values are very low 9%-16%. Even though it doesn't mean that your model is bad or worthless it’s worth discussing what it means (E.g. Maybe your model does not include all variables that are associated with the outcome)

The authors have focused on analysed data in the discussion section. However, the limitations and advantages of the study also needs to be included.

4. Conclusion

The last sentence – mentions optimizing physical health, while nowhere in the article physical health has been discussed. Please revisit.

Minor

In the 5th para – The second sentence is quite long, making it difficult to follow. Could you clarify what you mean? It would be helpful if you could break it down into several shorter sentences.

Under the results, in the title please correct brief COPE and HAM-A scores

The terms anxiety and coping scores and HAM-A and Brief COPE scores are used interchangeably throughout the manuscript. Suggest to use a uniform format.

Recreation drug users in the sample are quite high (36%). Is it a normal finding in the hospital set-up or a unique finding? Worthwhile to explain this to the international audience of this manuscript.

Tables need to be self-explanatory. Therefore, the symbols, short forms need to be described as a footnote

6. PLOS authors have the option to publish the peer review history of their article (what does this mean? ). If published, this will include your full peer review and any attached files.

**Do you want your identity to be public for this peer review?** For information about this choice, including consent withdrawal, please see our Privacy Policy .

Reviewer #1: No

Reviewer #2: No

---

## [Author Response · Author response to Decision Letter 0]

12 Feb 2025

Thank you for consideration of our manuscript, we have tried our effort best to improve the work according to the revisions requested by the journal. We have responded to the points mentioned by the reviewers under:

Line number and spacing has been added appropriately.

Major Revisions:

1. Introduction:

- We have added references to all mentioned instances as well as where we found were necessary and missing in the original document

- We have made an effort to acknowledge the sequencing in the paragraph and have structured in a way that shifts from broader to more local research, i.e. international, to any one specific country to Pakistan (country of study)

2. Materials and Methods:

- The final questionnaire was administered in Urdu, and we have now made this clear in the text as well as the reason why.

- The tools have been validated in Pakistan, appropriate references have been added.

- It is not the same population, the pre-test population was not included in the final sample, and this has been made clear now in the manuscript.

- The data collectors were authors not working at NICVD, not part of the health staff there. This point has been made clear now in the manuscript.

- Reason for assumption of normal distribution has now been elaborated in the manuscript.

3. Results:

- Ages were categorized to better reflect different stages of adult life, this has been further elaborated and explained now in the manuscript.

- The participants were divided based on income, explanation along with cut off values for different classes has been mentioned in the manuscript now.

- The tables have been heavily revised now, both to reduce redundancy and also to generally look neater and easier to interpret/read.

Discussion

- We have re-arranged the discussion in a way which elaborates our own findings before discussion on other literature<

- We have added discussion on the regression models.

- Limitations and advantages of the study have now been elaborated on.

4. Conclusion:

- The mentioning of physical health has been removed.

Minor Revisions:

- The mentioned sentence has been re-written to hopefully be more comprehensible.

- Corrected.

- Uniform format for HAM-A and brief COPE scores used.

- High incidence of recreational drug users has been elaborated on in results section now.

- Footnotes have been added.

---

## [Decision Letter · Decision Letter 1]

2 Mar 2025

PONE-D-24-52101R1Exploring anxiety symptoms, coping strategies, and socio-demographic influences among myocardial infarction patients in a tertiary care setting.PLOS ONE

Dear Dr. Ul Haq,

Thank you for submitting your manuscript to PLOS ONE. After careful consideration, we feel that it has merit but does not fully meet PLOS ONE’s publication criteria as it currently stands. Therefore, we invite you to submit a revised version of the manuscript that addresses the points raised during the review process.

We look forward to receiving your revised manuscript.

Kind regards,

Pasyodun Koralage Buddhika Mahesh

Academic Editor

PLOS ONE

Journal Requirements:

Reviewers' comments:

Reviewer's Responses to Questions

**Comments to the Author**

1. If the authors have adequately addressed your comments raised in a previous round of review and you feel that this manuscript is now acceptable for publication, you may indicate that here to bypass the “Comments to the Author” section, enter your conflict of interest statement in the “Confidential to Editor” section, and submit your "Accept" recommendation.

Reviewer #1: All comments have been addressed

Reviewer #2: (No Response)

2. Is the manuscript technically sound, and do the data support the conclusions?

Reviewer #1: Yes

Reviewer #2: Partly

3. Has the statistical analysis been performed appropriately and rigorously? 

Reviewer #1: I Don't Know

Reviewer #2: Yes

4. Have the authors made all data underlying the findings in their manuscript fully available?

Reviewer #1: Yes

Reviewer #2: Yes

5. Is the manuscript presented in an intelligible fashion and written in standard English?

Reviewer #1: Yes

Reviewer #2: Yes

6. Review Comments to the Author

Reviewer #1: (No Response)

Reviewer #2: The authors have significantly improved the manuscript with the recent amendments. However, a few minor issues still require attention and need to be addressed

Abstract

Line 48 – can re word to be more clear ‘e.g A cross-sectional study was conducted at the National Institute of Cardiovascular Disease (NICVD), Karachi, Pakistan, involving 203 patients diagnosed with myocardial infarction.

64-65 – suggest to remove physical health outcomes as it is not addressed in the current study

Introduction

Line 95 - The authors should clarify what the abbreviation HADS stands for.

Line 126 similar AKUADS

Line 143 to 148 – This sentence is very long, which makes its meaning confusing. Suggest to re word it

Line 148-152 Furthermore….. –It is unclear what point the authors are trying to make. Is it because it is a referral hospital or a specialized facility where patients from all over Pakistan visit?

Materials and Methods

164- Vicinity? The patients are from the NICVD. Vicinity gives the impression that the patients are from the surrounding area of the institute is it so?

171- To make the statement clear suggest adding ‘The questionnaire was first developed in English, and was translated into urdu….

192 – inclusion criteria

193 – where was the sample taken? different hospital

Results

239- it is customary to place the currency symbol at the beginning of the number, before the numerical value

In the tables, it is recommended to adopt a uniform format for decimal points. For example, in the first table authors use two decimal points, the second table uses one, and the 4th and 5th table alternates between two and three decimal points.

Table 8 – positive framing, last cell – alignment

References

Please pay more attention to references. There are a few errors. Include DOI/ Accessed date in appropriately. Suggest to recheck

e.g 625-627- reference 19 – not complete

re check references- 23, 24

7. PLOS authors have the option to publish the peer review history of their article (what does this mean? ). If published, this will include your full peer review and any attached files.

**Do you want your identity to be public for this peer review?** For information about this choice, including consent withdrawal, please see our Privacy Policy .

Reviewer #1: No

Reviewer #2: No

---

## [Author Response · Author response to Decision Letter 1]

7 Mar 2025

ABSTRACT

Thank you for your comments, your commitment to the betterment of the article is much appreciated!

- The sentence in question has now been reworded accordingly.

- The word in question has now been removed.

INTRODUCTION

Thank you for your comments, your commitment to the betterment of the article is much appreciated!

- Full form for both scales mentioned has now been included.

- The sentence in question has now been reworded to make it more comprehensible.

- The sentence in question was meant to emphasize how the study could be used as a reference to help map psychological states in myocardial infarction patients. Along with this it mentions how such info is potentially important for the prognosis of these patients. We realize mentioning the institution led to unnecessary confusion in the sentence and has subsequently been removed.

MATERIAL AND METHODS

Thank you for your comments, your commitment to the betterment of the article is much appreciated!

- The word in question has been removed to avoid confusion.

- The sentence in question has been updated accordingly.

- Inclusion criteria is mentioned in line 208.

- Clarity has been provided. The pilot was conducted among patients in the same institute however these patients were NOT included in the final sample.

RESULTS

Thank you for your comments, your commitment to the betterment of the article is much appreciated!

- Currency has now been mentioned before the number.

- We understand your recommendations. Although we have now adapted the 2nd table to have two decimal points. We feel the 3 decimal points (in 4th, 5th tables etc.) should remain the same, as changing this would have an effect on the accuracy of the results. There are several 3 decimal numbers in which the 2nd decimal point is “0” altering this into 2 decimals could make the results look widely different especially for variables such as p value. We also feel if we selectively alter the ones that don’t have 0 in their second decimal point it would appear untidy.

- Cell has now been aligned accordingly.

REFERENCES

Thank you for your comments, your commitment to the betterment of the article is much appreciated!

- DOI has been added where available. Where it was not available for any article, a direct link has been added instead.

- The mentioned reference has now been fixed.

- References have been revised.

---

## [Decision Letter · Decision Letter 2]

31 Mar 2025

PONE-D-24-52101R2Exploring anxiety symptoms, coping strategies, and socio-demographic influences among myocardial infarction patients in a tertiary care setting.PLOS ONE

Dear Dr. Ul Haq,

Thank you for submitting your manuscript to PLOS ONE. After careful consideration, we feel that it has merit but does not fully meet PLOS ONE’s publication criteria as it currently stands. Therefore, we invite you to submit a revised version of the manuscript that addresses the points raised during the review process.

We look forward to receiving your revised manuscript.

Kind regards,

Pasyodun Koralage Buddhika Mahesh

Academic Editor

PLOS ONE

**Journal Requirements:**

Reviewers' comments:

Reviewer's Responses to Questions

**Comments to the Author**

1. If the authors have adequately addressed your comments raised in a previous round of review and you feel that this manuscript is now acceptable for publication, you may indicate that here to bypass the “Comments to the Author” section, enter your conflict of interest statement in the “Confidential to Editor” section, and submit your "Accept" recommendation.

Reviewer #2: All comments have been addressed

Reviewer #3: All comments have been addressed

2. Is the manuscript technically sound, and do the data support the conclusions?

Reviewer #2: Yes

Reviewer #3: Yes

3. Has the statistical analysis been performed appropriately and rigorously? 

Reviewer #2: Yes

Reviewer #3: Yes

4. Have the authors made all data underlying the findings in their manuscript fully available?

Reviewer #2: Yes

Reviewer #3: No

5. Is the manuscript presented in an intelligible fashion and written in standard English?

Reviewer #2: Yes

Reviewer #3: Yes

6. Review Comments to the Author

**Reviewer #2:**  The authors have thoroughly addressed the queries raised during the review process, ensuring that all concerns and suggestions have been adequately considered. As a result, significant improvements have been made to the manuscript's quality, enhancing its clarity, coherence, and overall presentation. The revisions reflect the authors' commitment to producing a well-refined and comprehensive document.

**Reviewer #3: ** I would like to acknowledge the authors for their valuable contributions to this study. Their hard work and dedication have been integral to the success of this research. The reviewer comments have been addressed and corrected; however, there are some minor changes that still need to be addressed.

92 -treatment concerns, lack of support etc [4]. Studies in the past have highlighted the 93 prevalence of anxiety in the hospital setting-Please mention whether these patients were admitted with a life-threatening condition or for any other reason.

115- significantly higher risks for anxiety (HR = 5.06) and depression (HR = 7.23 ) Please mention HR is refers to Hazard Ratios (HR),

188 The questionnaire was given to the data collectors (authors of the study who did not 189 currently work at NICVD)-Could you clarify whether you're distributing the questionnaire within a specific time after admission in the study, or is it the time period given for participants to complete it

221- while 69 (34.0%) were female participants. Average age of the sample population was 54- hope this is mean ,if so please mention

547 In conclusion, this study details the significant prevalence of anxiety among myocardial infarction patients and highlights the critical role of coping strategies in their psychological well-being. How do you say this it is not mention in the results ?? it should be significant associated factors among 203 patients

7. PLOS authors have the option to publish the peer review history of their article (what does this mean? ). If published, this will include your full peer review and any attached files.

**Do you want your identity to be public for this peer review?** For information about this choice, including consent withdrawal, please see our Privacy Policy .

Reviewer #2: No

Reviewer #3: **Yes: ** W.D.J.K Amarasena

---

## [Author Response · Author response to Decision Letter 2]

1 Apr 2025

Comment 1: Please mention whether these patients were admitted with a life-threatening condition or for any other reason.

Author response: Thank you for your comments! We have now elaborated more on the setting; i.e. newly admitted acute care patients.

Comment 2: 115- significantly higher risks for anxiety (HR = 5.06) and depression (HR = 7.23 ) Please mention HR is refers to Hazard Ratios (HR)

Author response: Thank you for your comments! We have now included in the full form of HR (i.e. Hazard ratio).

Comment 3: 188 The questionnaire was given to the data collectors (authors of the study who did not 189 currently work at NICVD)-Could you clarify whether you're distributing the questionnaire within a specific time after admission in the study, or is it the time period given for participants to complete it

Authors response: Thank you for your comments! We have now elaborated further on when the questionnaire was distributed (i.e. on recruitment).

Comment 4: 221- while 69 (34.0%) were female participants. Average age of the sample population was 54- hope this is mean ,if so please mention

Author response: Thank you for your comments! Yes, it was Mean, and we have now corrected it.

Comment 5: 547 In conclusion, this study details the significant prevalence of anxiety among myocardial infarction patients and highlights the critical role of coping strategies in their psychological well-being. How do you say this it is not mention in the results ?? it should be significant associated factors among 203 patients

Author response: Thank you for your comments! We have reworded the aforementioned sentence so that it is more accurate to our study.

---

## [Decision Letter · Decision Letter 3]

16 Apr 2025

Exploring anxiety symptoms, coping strategies, and socio-demographic influences among myocardial infarction patients in a tertiary care setting.

PONE-D-24-52101R3

Dear Dr. Ul Haq,

We’re pleased to inform you that your manuscript has been judged scientifically suitable for publication and will be formally accepted for publication once it meets all outstanding technical requirements.

Kind regards,

Pasyodun Koralage Buddhika Mahesh

Academic Editor

PLOS ONE

Additional Editor Comments (optional):

Reviewers' comments:

Reviewer's Responses to Questions

**Comments to the Author**

1. If the authors have adequately addressed your comments raised in a previous round of review and you feel that this manuscript is now acceptable for publication, you may indicate that here to bypass the “Comments to the Author” section, enter your conflict of interest statement in the “Confidential to Editor” section, and submit your "Accept" recommendation.

Reviewer #3: All comments have been addressed

2. Is the manuscript technically sound, and do the data support the conclusions?

Reviewer #3: Yes

3. Has the statistical analysis been performed appropriately and rigorously? 

Reviewer #3: Yes

4. Have the authors made all data underlying the findings in their manuscript fully available?

Reviewer #3: Yes

5. Is the manuscript presented in an intelligible fashion and written in standard English?

Reviewer #3: Yes

6. Review Comments to the Author

Reviewer #3: Thank you for your hard work in revising the manuscript according to my comments. I appreciate your efforts and the improvements made.

7. PLOS authors have the option to publish the peer review history of their article (what does this mean? ). If published, this will include your full peer review and any attached files.

**Do you want your identity to be public for this peer review?** For information about this choice, including consent withdrawal, please see our Privacy Policy .

Reviewer #3: **Yes: ** W.D.J.K Amarasena

---

## [Editor Report · Acceptance letter]

PONE-D-24-52101R3

PLOS ONE

Dear Dr. Ul Haq,

I'm pleased to inform you that your manuscript has been deemed suitable for publication in PLOS ONE. Congratulations! Your manuscript is now being handed over to our production team.

Kind regards,

on behalf of

Dr. Pasyodun Koralage Buddhika Mahesh

Academic Editor

PLOS ONE